

# Uncertainty and detectability of climate surface response to large volcanic eruptions

Fabian Wunderlich[1,2] and Daniel M. Mitchell[1]

[1]Atmospheric, Oceanic and Planetary Physics, University of Oxford, UK
[2]Now at Institute of Meteorology, Freie Universität Berlin, Germany

*Correspondence to:* F. Wunderlich, Institute of Meteorology, Freie Universität Berlin, Carl-Heinrich-Becker-Weg 6-10, 12165 Berlin, Germany. (fabian.wunderlich@met.fu-berlin.de)

**Abstract.** In light of the range in presently available observational, reanalysis and model data, we revisit the surface climate response to large tropical volcanic eruptions from the end of the 19th century until present. We focus on the dynamical driven response of the North Atlantic Oscillation (NAO) and the radiative driven tropical temperature response. Using ten different reanalysis products and the Hadley Centre Sea Level Pressure observational dataset (HadSLP2) we confirm a positive tendency

in the phase of the NAO during boreal winters following large volcanic eruptions, although conclude that it is not as clear cut as the current literature suggests. Especially during poorly observed periods where higher uncertainties produce a less robust signal. The phase of the NAO leads to a dynamically driven warm anomaly over Northern Europe. At the same time, there is a general cooling of the tropical surface temperatures due to the reduced incoming shortwave radiation. The magnitude of this cooling is uncertain and is hard to isolate using observational data alone (mainly due to the presence of El Niño). Therefore

we use regression-based detection and attribution techniques to investigate the volcanic temperature signal with eight Coupled Model Inter-comparison Project phase 5 (CMIP5) models. In all models the volcanic signal can be detected but a general overestimation of the surface cooling is found. The enhanced surface cooling in models is likely driven, in part, by an over absorption of SW radiation in the lower stratosphere, but aliasing with El Niño events is also an issue and further process based studies are necessary to confirm these.

## 1 Introduction

Understanding of the atmospheric naturally-forced variability is a key issue to estimate the human induced contribution to the recent climate change. Large volcanic eruptions can have an impact on the climate over several years (Robock, 2000). The injected aerosols into the lower stratosphere influence the radiation balance, resulting in a cooling of the tropical surface temperature (Humphreys, 1913, 1940) and a heating in the tropical lower stratosphere (Labitzke and McCormick, 1982; Parker and Brownscombe, 1983). For the eruption of Mt. Pinatubo in June 1991, which was the strongest tropical eruption in the satellite

era, the lower tropical stratosphere was warmed up to 3 K (Mitchell et al., 2014; Fujiwara et al., 2015). The cooling signal on the surface is less pronounced and therefore more difficult to separate from other internal and external climate variability. To perform this separation Mitchell et al. (2014) and Fujiwara et al. (2015) used multiple linear regression with nine reanalysis datasets (ERA-Interim, ERA-40, JRA-25, JRA-55, MERRA, NCEP-R1, NCEP-CFSR, NCEP-R2 and NCEP-20CR). A good



agreement between most reanalysis datasets for the temperature response in the stratosphere was found. Uncertainties which arise from variability of the differences between the reanalysis models are currently being assessed in the Stratospheric and tropospheric Processes And their Role in Climate - Model inter comparison Project (SPARC-RIP; (Fujiwara and Jackson, 2013)). The agreement between the reanalysis datasets in the troposphere is strong but no clear tropospheric cooling could be found,

taking into account all large tropical eruptions after 1960 (Fujiwara et al., 2015).

The purpose of this study is two fold. Firstly we examine the dynamical driven North Atlantic Oscillation (NAO) variability following volcanic eruptions, including the uncertainty in observations and reanalysis to assess if the signal is robust, given newly available data sets. We then consider the radiative response for the same reason, but using detection and attribution techniques. The remainder of this paper is as follows. Datasets and methods used in this study are described in section 2,

followed by the results in section 3 and the discussion and summery in section 4.

## 2   Data and analysis method

In this study we use near surface monthly mean Temperature of Air at the Surface (TAS) and sea level pressure data from ten available reanalyses (Table 1). Furthermore we analyse the TAS fields of the CMIP5 historical model experiments provided by the World Climate Research Programme (Taylor et al., 2012).

For comparison of the temperature we use the Met Office Hadley Centre HadCRUT4 dataset, available with 100 ensemble members covering the period 1850 until present (Morice et al., 2012). This product has a global coverage with a resolution of $5°x5°$ but it includes missing data depending on the observational data base. The provided data are anomalies referring to the mean climatology between 1961-1990 based on the relatively small number of missing data during this period.

To compare with sea level pressure data we make use of the Hadley Centre Sea Level Pressure dataset (HadSLP2) (Allan and

Ansell, 2006). This product does not contain missing data because of an applied interpolation procedure, which can cause uncertainties, specially in less covered regions like the Arctic, Antarctic or deserts. The spatial resolution of $5°x5°$ is equal to the temperature product but the data includes the period 1850 until 2004. However, an updated version is available using NCEP/NCAR reanalysis fields (Kalnay et al., 1996) from 2005 until present, named HadSLP2r. The mean values for both dataset are homogeneous but the variance is higher in HadSLP2r. Nevertheless we consider this adding of the dataset as

justified since we use the updated period only for the calculation of the climatology, significance testing and for the Empirical Orthogonal Function (EOF) analysis in order to calculate the NAO time series (Thompson and Wallace, 1998).

### 2.1   Reanalysis Data

All reanalysis datasets span at least the period from 1979 until 2012 except two ECMWF products: ERA-20C (20th century reanalysis product) (Poli et al., 2013), which ends in December 2010 and ERA-40 (Uppala et al., 2005), which ends in August

2002. To be able to compare all reanalysis over the same temporal region, we extend the ERA20C and ERA-40 datasets until 2012 with data of the ERA-Interim reanalysis. ERA-Interim provides some advantages to the ERA-40 version including an improved data assimilation and a better representation of the stratospheric circulation (Dee et al., 2011). Since we use this




added data just indirectly for calculations of climatology, anomaly fields and the EOF, the differences which would arise by using a full period consideration are negligible.

Both reanalysis products of the Japan Meteorological Agency (JMA) are used for the study. JRA-25 ends in 2004 but the data until present is available from the JMA Climate Data Assimilation System (JCDAS) with the same system as JRA-25

(Onogi et al., 2007). The subsequent JMA product is called JRA-55 and covers a longer period beginning from 1958. Several improvements have made in comparison to the previous product such as a significant reduction of the large temperature bias in the lower stratosphere by using a new radiation scheme (Ebita et al., 2011).

The MERRA reanalysis obtained from the National Aeronautics and Space Administration (NASA) is focused on the correct simulation of the hydrological cycle and is the only reanalysis used which does not represent the analysis field with spectral

coefficients (Rienecker et al., 2011).

The first reanalysis project was operated by the National Centers for Environmental Prediction (NCEP), called NCEP-R1 (Kalnay et al., 1996) and has been updated with the NCEP-R2 product (Kanamitsu et al., 2002). A more sophisticated and recent reanalysis product of NCEP is the Climate Forecast System Reanalysis (NCEP-CFSR) (Saha et al., 2010).

Most products include a period beginning from the late 20th century, which means that just the last two large tropical eruptions

of Mt. Pinatubo and El Chichón can be considered for the analysis. Due to the large number of individual ensemble members and the long time series beginning in 1871, the NOAA-CIRES 20th Century Reanalysis (NCEP-20CR) is the main product used in this study (Compo et al., 2011). The assimilation scheme of the NCEP-20CR reanalysis product uses a ensemble Kalman Filter in streams of 5 years. Each stream has 56 members.

The ERA-20C product starts in 1900 and includes most of the period which we investigated. The assimilation of these long

datasets only include surface observation data, in contrast to the other products assimilating also satellite and radio-sonde measurements. The reanalysis datasets are generally in good agreement with surface observation data, specially for sea level pressure and near surface temperature data, used in this study (Simmons et al., 2004; Makshtas et al., 2007; Lindsay et al., 2014). A warm bias of the arctic winter temperature in the NCEP-20CR dataset due to less sea ice concentration at coastal regions was reported by Compo et al. (2011). NCEP-R1 and JRA-25 show differences in the sea level pressure field over Green-

land and MERRA generally over mountain areas likely because of distinguish surface pressure reduction methods (Lindsay et al., 2014). Since we use just anomaly fields for our calculations, this should not affect the results significantly.

## 2.2 CMIP5 models

The model data are obtained from the historic simulations of the CMIP5 models which include simulations with just volcanic forcing (Table 2). For the detection and attribution analysis we use the simulations with separately forced anthropogenic,

solar and volcanic forcing. The Mark 3.6 model of the Commonwealth Scientific et Industrial Research Organization (CSIRO-Mk3.6) does not include a single simulation for the solar forcing (Rotstayn et al., 2010). Therefore we subtract the volcanic simulation from the simulation with just natural forcing, in order to receive a solar-only signal. All models except the coupled-physical model of the Geophysical Fluid Dynamics Laboratory (GFDL-CM3) start from the beginning of 1850 and have at least three ensemble members. The advantages of the GFDL-CM3 model is a sophisticated interaction scheme between



aerosols and clouds and a focus on coupling between the troposphere and stratosphere (Donner et al., 2011). The Community Climate System Model 4 (CCSM4) and the Community Earth System Model version 1 with Community Atmospheric Model version 5 (CESM1-CAM5) models use the aerosol optical depths description of Ammann et al. (2007). All other models use the updated version of the Sato et al. (1993) description. CESM1-CAM5 includes the direct and indirect effects of aerosols (Meehl et al., 2013), while CCSM4 just provides the direct effects. These models show a good reproduction of the ENSO due to an improved deep convection scheme in the atmosphere component (Gent et al., 2011).

The ModelE2 version of the NASA Goddard Institute for Space Studies (GISS-E2) provides four different simulations with just volcanic forcing (Schmidt et al., 2014). They differ by using distinguished ocean models and whether the models include interactive chemistry and parametrization of indirect aerosol effects. GISS-E2-R uses the Russell ocean model (Hansen et al., 2007) and GISS-E2-H uses the Hybrid Coordinate Ocean Model (HYCOM) (Sun and Bleck, 2006). Both realizations are available in a version with non-interactive chemistry (NINT), comparable to the prior CMIP3 simulation, but with a tuned aerosol indirect effect following Hansen et al. (2005) and a version with Tracers of Chemistry, Aerosols and their Direct and Indirect effects (TCADI) including interactive chemistry and a parametrization of the first indirect aerosol effects (Menon et al., 2010).

## 2.3 Choice of volcanoes

Not all studies which concentrate on the large-scale impact of volcanic eruptions use the same criteria for the choice of which eruption should be considered for a composite analysis. The volcanic explosivity index (VEI) introduced by Newhall and Self (1982) is a very frequently used measurement for the strength of the eruption (Robock, 2000). The calculation of the index is restricted to volcanic measurements but does account for the height of the eruption column. In Figure 1 all volcanic eruptions since the 19th century with a VEI-index of at least 5 are shown. Additionally volcanoes are included which in other studies are considered to have an impact on the climate but just reached a VEI-index of 4. The size of the triangle of each eruption indicate the respective VEI-index. The colour show the phase of the El Niño-Southern Oscillation (ENSO) in the first winter after the eruption. The last three tropical eruptions of Mt. Agung in 1963, El Chichón in 1982 and Mt. Pinatubo in 1991 were followed by an El Niño event, suggesting a possible connection between large tropical volcanic eruptions and ENSO (Adams et al., 2003). It may also be that the events are coincidental, but either way, it makes the analysis harder to interpret.

Climate models represent volcanic eruption by an increase of the atmospheric aerosols due to the ejected material. Most models use the updated version of the so called Sato-Index (Sato et al., 1993). This index shows the aerosol optical depth (AOD) at wavelength 550 nm and is available as a zonal mean with global coverage and a meridional resolution of around 8°. In Figure 2 the tropical (30°S-30°N) AOD is plotted. As expected from the chosen region the values of low latitude eruptions are generally more pronounced than the extratropical eruptions. The tropical region is characterised by rising air in the stratosphere which lift the aerosols into higher levels. The residual stratospheric meridional circulation transports the aerosols to high latitudes (Trepte and Hitchman, 1992). A volcanic eruption in higher latitudes is expected to have less influence on the climate system because the downward flow in the stratospheric extratropics avoids rising aerosols in higher levels. Nevertheless some studies could show that also extratropical volcanic eruptions can have a significant large scale impact on climate but usually just on



the hemisphere where the eruption took place (Graf and Timmreck, 2001; Oman et al., 2005). Since we focus on both the particular impact of the eruption on the NAO and the global temperature response, we consider the AOD in the tropical middle and upper stratosphere. In Figure 2 we draw a threshold at 0.05 nm in order to distinguish which volcanoes likely have a strong impact on the global climate. Therefore we choose the eruption of Krakatau in August 1883, Santa María in October 1902, Mt.

Agung in March 1963, El Chichón in April 1982 and Mt. Pinatubo in June 1991 for our analysis.

## 2.4 Methods

For the calculation of the NAO we use the leading EOF between $0°$-$70°$W and $35°$N-$80°$ during the period 1979-2012 (Thompson and Wallace, 1998; Baldwin et al., 2008). We consider the SLP anomalies of the two winters following the eruption as volcanic influenced, except in the case of Santa María which erupted so late during the year that the full influence on the winter

circulation is unlikely. The anomalies are calculated with respect to the mean for the years 1979-2012, excluding the following two years after the eruptions of El Chichón and Mt. Pinatubo.

The analysis of the temperature fields will be compared to the HadCRUT4 dataset, consisting of anomalies relative to the 1961-1990 reference period (Morice et al., 2012). As some reanalysis products start in 1979 we just consider the five reanalysis datasets which include this reference period (ERA-20C, ERA-40, JRA-55, NCEP-R1, and NCEP-20CR). After removing

the mean seasonal cycle we subtract the data with a 10 year running mean to remove any further trend. To make sure that the running mean is not influenced by the considered year, we average over the 60 months before the year and the 60 months after the considered year. To investigate the volcanic influence on the temperature field we average over the first and the second twelve months following the eruption.

The TAS preprocessing of the CMIP5 model means for the detection and attribution analysis is equal to the missing value

consideration in Jones et al. (2013). In this study we use means over the tropical region from $30°$S to $30°$N , averaged over a period of three years in order to minimize a possible bias towards ENSO signals.

By using models with just single forcings (e.g. anthropogenic or volcanic), a characteristic response pattern, called fingerprint can be found (Hasselmann, 1979, 1993). A frequently used assumption is, that every fingerprint can be combined in a linear regression model in order to best fit the observations (Allen and Tett, 1999). By taking uncertainties of the observations and the

model simulated response into account, the total least square algorithm (TLS) is applied to minimise the orthogonal distance to the best fit (Allen and Stott, 2003). The internal climate variability must be derived from control runs of the model or inter-ensemble differences from the ensemble mean in order to estimate the climate noise covariance (pseudo-observations). We use both derivations and furthermore overlap noise segments by shifting the periods by 10 years. Therefore we can maximise the number of pseudo-observations.

Due to the high amount of data, the calculation of the noise covariance normally demands a reduction of the dimensionality. Most studies calculate an EOF based pseudo-inverse with the disadvantage that the number of EOF's is limited and the result depends on the chosen truncation. Ribes et al. (2009, 2013) propose a solution of this issue by introducing the Regularized Optimal Fingerprint (ROF) technique with a well-conditioned estimator of the covariance matrix. Using this technique the resulting regression coefficients, called scaling factors are not dependent on the truncation. Each forcing corresponds to the





best fit amplitude of the simulated signals. A signal is detected in the observations if the scaling factor and 5-95% uncertainty range is different from zero. A detected signal which does not include a scaling factor of one is either overestimated, when the factor is between zero and one or underestimated, when the factor is bigger than one (e.g., Hegerl and Zwiers, 2011). The uncertainty of the scaling factors is determined with a residual consistency test (RCT) using Monte Carlo simulation (Ribes
et al., 2013).

## 3   Results

### 3.1   Pressure and NAO response

To analyse the NAO response to large volcanic eruptions we use surface pressure data of all available reanalysis products. Since MERRA, ERA-Interim, JRA25, NCEP-R2 and NCEP-CFSR start in 1979 with the beginning of continuous satellite
observations we use the years 1979-2012 as our reference period. Two tropical eruptions had a significant impact on the climate system during this period, El Chichón in 1982 and Mt. Pinatubo in 1991. Figure 3 shows the mean SLP anomaly in the first two post volcanic winters after El Chichón and Mt. Pinatubo over the extratropical Northern Hemisphere in observation and for the multi-reanalysis mean. The significance is calculated with a Monte Carlo test assuming independence between the volcanic eruption events.

The response pattern is captured well in all reanalysis products. Since the assimilation of surface pressure data is essential for reanalysis products, the difference between the individual products and the observations is expected to be small (Kalnay et al., 1996). Over the Arctic region a higher level of observational uncertainty is apparent due to the decreased number of assimilated measurements than in the middle latitudes. Low pressure anomalies over the Arctic region are observed, bordered by a positive signal with the centre over Europe. Also negative anomalies can be seen over the North Pacific, likely due to the sampling of
El Niño events or due to increased wave reflection during winters with a strong polar vortex (Perlwitz and Graf, 2001).

Differences between the reanalysis products are weak but can be seen mainly above mountain areas, e.g. Rocky Mountains (Supplementary Information, Figures S1,S2), likely due to different pressure reduction techniques Lindsay et al. (2014). The NCEP-20CR reanalysis product is in good agreement. NCEP-20CR is found to capture well the stratospheric temperature response to volcanic eruptions but with a slightly lower amplitude in comparison to other reanalysis products (Mitchell et al.,
2014; Fujiwara et al., 2015). At the surface the NCEP-20CR does not show major differences of the pressure response after the eruption of Mt. Pinatubo and El Chichón (Figures S1,S2).

In comparison to an expected positive NAO, in the first winter the high SLP anomaly is shifted towards Central Europe and therefore the negative centre is shifted northwards (Hurrell and Deser, 2009; Hurrell, 1995). Fujiwara et al. (2015) showed that the eruption of Mt. Pinatubo influences the stratospheric temperature and circulation stronger than the eruption of El Chichón.
Therefore it is expected that the surface pressure signal is dominated by the response to the eruption of Mt. Pinatubo. Probably due to the east phase of the quasi-biennial oscillation (QBO) in the first winter after the eruption of Mt. Pinatubo the volcanic signal on the SLP was weakened (Stenchikov et al., 2002). In contrast, the response of the second winter (Figure 3 b,d) shows a more NAO like pattern over the North Atlantic with significant positive anomalies in the region of the Azores. A sample of





only two single events is not a robust data basis to make general conclusions. Hence, we use the NCEP-20CR product which agrees well with the other reanalysis products, considering the SLP response after the eruption of Mt. Pinatubo and El Chichón (Figures S1,S2) and contains a longer period between 1871-2012. During this period five major tropical eruptions took place: Krakatau, Santa María, Mt. Agung, El Chichón and Mt. Pinatubo. The mean SLP response to these five eruptions is shown in

Figure 4. The pattern in the North Atlantic region is similar to the mean over the last two eruptions but the pressure anomaly gradient is much smaller, with a predominantly significant response pattern in the first winter. In general the average pressure anomaly field indciates a shift towards a positive NAO in the first winter after the eruption. A significant mean NAO signal could be found by Christiansen (2008) and D12, taking into account tropical and extratropical eruptions. We calculated the monthly winter NAO index (see Methods) of just the tropical eruptions and show the index individually for every eruption in

Figure 5. The observation data is shown by crosses and all reanalysis data is shown by blue ranges. The orange ranges indicate the NAO index of the 56 NCEP-20CR ensemble members (95% ensemble spread).

The spread of the reanalysis products is small showing high agreement between the individual reanalysis products. In most cases the observations agree with the reanalysis data but in some cases the reanalysis NAO response can differ from the observational signal (e.g. El Chichón eruption in December and January). Especially with higher uncertainty of the observation

data in the post satellite era, the differences of the NAO index to the reanalysis data are bigger and the error bars of the ensemble spread are generally wider. Therefore some ensemble members can differ with the corresponding observations by strength and even sign of the NAO phase (e.g. Santa María in December, Mt. Agung in February).

Consistent with Jones et al. (2003) there is no evidence for a positive NAO shift due to the volcanic eruptions in particular months and in the second winter after the eruption (Figures S3). The response in the mean winter NAO is not clear. In the first

winter after the three eruptions of Krakatau, El Chichón and Mt. Pinatubo a clear positive NAO phase could be found. For the eruption of Santa María we do not use the first winter after the eruption because the volcano erupted so late during the year that an influence of the injected aerosols to the stratospheric circulation is unlikely. In the winter 1903/04 a NAO-index around zero was found. This agrees with the results of Christiansen (2008) and D12. In the winter directly after the eruption (1902/03) a strong positive NAO was found (Christiansen, 2008). In the winter after the eruption of Mt. Agung we found a negative NAO.

Most of the aerosols after the eruption of Mt. Agung were concentrated in the Southern Hemisphere (Figure 2), which reduces the impact on the boreal stratosphere in winter. Therefore we conclude that we do not find a significant positive NAO response to volcanic eruptions with taking just the strongest five tropical eruptions from the end of the 19th century until present. We confirm that the NAO generally shifts towards a positive state in Figure 4 and in Figure 5d) but exceptions like Mt. Agung or Fernandina are found (Figure S4). Responsible for these exceptions can be disturbances of the polar vortex which descend

downwards and influence the NAO over a short time scale of some weeks (Baldwin and Dunkerton, 1999). The positive NAO response to volcanic eruptions leads to a positive temperature anomaly over Northern Europe in winter (Robock and Mao, 1992, 1995; Fischer et al., 2007). This warming disagrees with the radiative driven cooling of the troposphere following the eruptions. Therefore just dynamically driven effects like circulation changes could explain this response. It is important to understand these dynamical driven effects in order to understand the total volcanic signal. The studies of Stenchikov et al.

(2006) and D12 could show that general circulation models are generally not able to reproduce this secondary effects and





hence it is questionable if they reproduce the temperature response well. Only a subset of models show an associated warming over Northern Eurasia but much weaker than the observations (D12; Gillett and Fyfe, 2013).

## 3.2 Temperature response

The TAS anomaly composites over the last three large volcanic eruptions of the reanalysis products in comparison to the observation data are shown in Figure 6. We note here that, due to the observational anomalies respective to the period 1961-1990, we only consider the five reanalysis products which include this period. To remove a further trend we subtracted an adopted running mean (see Methods). We find a significant warming over Northern Europe which is consistent to the expected winter warming over this region. The warming over Siberia and the cooling in the Middle East suggests a positive NAO response following volcanic eruptions (Thompson and Wallace, 1998). The significant temperature response in the first year after the eruption over the pacific region is very similar to the temperature pattern during EL Niño events (Deng et al., 2012), hence it is likely a cause of the sampling of positive ENSO phases following all three eruptions (Stenchikov et al., 2006). The expected global cooling in the first years after the eruption is not obviously due to the overwhelming ENSO warming over the Pacific. An early study of (Robock and Mao, 1995) shows a general cooling by removing the ENSO signal. A predominant but generally not significant cooling is present especially in the second year after the eruption, when the signal is much less influenced by ENSO events. To assess a larger sample of eruption events we expand the considered period and include the early eruptions of Krakatau and Santa María. During the period after this eruptions much less observation data is available, especially over continental areas. In Figure 7a,b we averaged over at least four of five years after the volcanic eruptions. The results are similar to those of Figure 6. To be able to evaluate the effect of the volcanic eruptions over less observed areas we calculated the mean temperature response for all five eruptions with the NCEP-20CR product without missing data consideration (Figure 7c,d). Also the consideration of five independent eruptions confirms the significant warming over Northern Europe due to the shift of the NAO towards a positive state. The response of the second year after the eruptions, with almost absent ENSO influences shows a significant cooling over the Central Atlantic and over the Western Pacific. In the tropics a general cooling is found over land and sea areas with exception over the Eastern Pacific with minor and not significant warming. This would confirm that the cooling after large volcanic eruptions is strongest in the tropics [D12]. For this reason we focus in the following on the TAS signal in the tropical region (30°N-30°S).

Figure 8 shows the mean anomalies of the tropical temperature in the first and second year after the eruptions, relative to the climatology of 1961-1990 (with removed trend). The reanalysis and observation data are generally in good agreement, except after the eruption of Krakatau, with a significant separation of the anomaly spreads. Relative to the sparsity of observations in the late 19th century these differences are reasonable. A general cooling of the tropical TAS response is not clearly visible in the observation and reanalysis data. After the eruption of El Chichón just positive anomalies in both post volcanic years are found. Also the study of Fujiwara et al. (2015) did not show a clear temperature signal in the tropical troposphere after the eruption of El Chichón. The eruption of Mt. Pinatubo was the strongest in the 20th century and caused a cooling around 0.1 to 0.2 K, according to the observation and reanalysis data. The historical simulations of the CMIP5 models involve far higher range of TAS responses than the observations. Most simulations with just volcanic forcing show a negative temperature signal.





For consistency we calculated the anomalies relative to the same period as the observation and reanalysis data. This period (1961-1990) was highly influenced by volcanic activity. By calculating anomalies relative to a non-volcanic influenced period (e.g. 1931-1960) we found that all simulations with just volcanic forcing show a negative TAS response after both years of all five eruptions (not shown). The TAS response to the eruptions of the historical simulations including also anthropogenic

and other natural forcing agrees well with the simulation with just volcanic forcing. In most cases the mean response of the historical simulations (indicated by the dot between the whiskers) is colder than the TAS signal of the observation or reanalysis data. But due to the large spread of simulated TAS response, the results are not statistically significant. Nevertheless a tendency for the models to simulate a colder volcanic response than observed may be supposed (Eyring et al., 2006). Since we did not apply a regression technique to the observation and reanalysis data, we can not assume that the TAS response is not influenced

by anthropogenic or other natural forcings, including internal variability such as the El Nino signal. Instead we use detection and attribution techniques to regress the volcanic forcing directly onto the observed temperature (Allen and Tett, 1999; Stott et al., 2003). With this method we separate the volcanic signal directly from other forcings.

## 3.3 Detection and Attribution

Figure 9 shows the time series of three-yearly averaged, separately forced CMIP5 simulations of TAS anomalies averaged over

the tropical region (30°N-30°S) along with the observations. A characteristic increase of the TAS anomalies in the observational record and the anthropogenic historical simulations of the CMIP5 models is shown (black and blue lines, respectively), The volcanic simulations indicate no dominant trend but with strong negative signals following all five volcanic eruptions, whereas the solar simulations indicate a weak signal around the zero line. The results do not differ essentially from previous studies using global averages instead of tropical (Jones et al., 2013; Hegerl and Zwiers, 2011), although here the solar and

volcanic signals are seperated. The TAS anomaly differences between the models will not be discussed in detail but all model time series can be found in the Supplementary Information (Figure S6). The strong negative temperature bias in the versions with Tracers of Chemistry, Aerosols and their Direct and Indirect effects (TCADI) of the GISS models after the eruption of Mt. Agung could be a cause of a different consideration of the chemical response to volcanic eruptions. The consideration of interactive chemistry could lead to feedback processes which cause a longer influence of the ejected material to the atmosphere.

We applied the ROF technique to a three forcing analysis using the anthropogenic, solar and volcanic simulations. Results were sporadic, and this is because the solar signal is very weak, and leads to the regression model not fitting all relevant forced response patterns correctly. As such, we choose to exclude the solar signal from our ROF analysis. A signal-to-noise calculation on the solar only simulations is low, and therefore we conclude that the solar signal on surface tropical climate is not detected (in agreement with (Stott et al., 2003)). We therefore continue using a two-signal ROF analysis of anthropogenic and volcanic

forcing. Note that including the solar and volcanic signal into one 'natural' forcing yields very similar results. Figure 10 shows the scaling factors, their uncertainties and the residual consistency test (RCT) of anthropogenic and volcanic forcings of all individual models and the multimodel mean.





With the exception of CSIRO-Mk3.6, all models and the multimodel mean have detectable volcanic signals. Three models and the multimodel mean show a significant overestimation of the volcanic signal and none of the models show an underestimation. Ribes and Terray (2013) used the natural forcing instead of just the volcanic and provided very similar results. Since we use just the tropical region, the sampling of El Niño events could be biased by a warming over the Pacific. The variability

due to ENSO phases is included in our control runs and intra-ensemble mean differences, which are used in the ROF analysis. When we provide the same analysis for the global coverage (instead of only 30°S-30°N) every model has a detectable volcanic signal and almost all models show an overestimation of the volcanic signal with a mean scaling factor best-estimate of about 0.5 (Figure S7). This suggests that a sampling of El Niño events is only partially responsible for the overestimation of the surface volcanic signal. It is also the case that the models have too large temperature response to volcanic eruptions in the lower

tropical stratosphere (Mitchell, 2015; Mitchell et al., 2015), suggesting that more solar radiation is absorbed in this region, and therefore less gets to the surface (and hence further cooling in the models).

The results for the anthropogenic forcing show that most models underestimate and some overestimate the anthropogenic influence on the tropical surface temperature. The mean response shows a small underestimation of the anthropogenic signal in the model. The underestimation of the anthropogenic signal of both GISS-TCADI models might be influenced by the cold bias

of the temperature after the Mt. Agung eruption. This cold bias counteracts the warm bias due to anthropogenic forcing, which makes an enhanced warming necessary in order to best fit the observations.

To represent the best fit forcing patterns, consistent with observations, the mean volcanic and anthropogenic signals are multiplied by their corresponding scaling factor (Figure 11). Focusing on the volcanic signal, there is a lower mean TAS response than expected from Figure 9. The mean cooling of around 0.1 K is consistent to the results of the reanalysis and observation

data in Figure 8. Studies which evaluate the entire natural forcing show a strong cooling after strong volcanic eruptions (e.g., Jones et al., 2013). Therefore also the TAS signal due to natural forcing could be overestimated by the models (Ribes and Terray, 2013). The anthropogenic forcing signal follows the warming of the observational record in tendency and strength quite well and is consistent to the results found in the last IPCC report (Stocker et al., 2013).

## 4 Conclusion and Discussion

In this study we investigated the uncertainty in surface climate response to strong volcanic eruptions. The most up-to-date available reanalysis products, GCMs and the newest observation datasets are used to best evaluate the radiative driven tropical temperature response and the dynamical driven NAO response following eruptions. Given the availability of these new datasets, it is timely to revisit the surface response to test the robustness of past studies of volcanic influences on climate. A summary of volcanic eruption intensity, and occurrence of El Niño events is present in Figure 1 from 1880-present.

The shift of the NAO towards a positive state in boreal winter due to an intensification of the polar vortex was noted in some observational studies (e.g., Shindell et al., 2004) but only some models with a good representation of the stratosphere are able to reproduce the associated winter warming over Eurasia (Kirchner et al., 1999). The CMIP5 models generally fail or underestimate the impact of the volcanic eruptions on the Northern Hemispheric circulation [D12]. This shows that the



dynamical mechanism is still not fully understood (Graf et al., 2007). Conditional on the injected material into the stratosphere, we selected the strongest five tropical volcanic eruptions from the end of the 19th century until present for our analysis. They are expected to have the biggest impact on the atmosphere.

We showed that a positive NAO phase is likely to be present during the first post volcanic winter, but uncertainties still remain

because not all winters following large tropical volcanic eruptions show a positive NAO (Mt. Agung and Fernandina). Also none of the particular winter months show a significant shift of the NAO after the eruptions. By taking into account all available reanalysis datasets and the HadSLP2 observation data we have seen that there is a general agreement between the datasets. It is known that the atmospheric condition after the volcanic eruption is a big source of uncertainty for the impact of eruptions on the NAO. The QBO phase (Holton and Tan, 1982; Stenchikov et al., 2004), an El Niño or La Niña event (Moron and Plaut, 2003;

Manzini et al., 2006; García-Herrera et al., 2006; Calvo et al., 2009) as well as the solar variability (Lean et al., 1995; Haigh, 2002; Gray et al., 2013) can influence the NAO phase directly or indirectly by modulating the stratospheric winter circulation in the Northern Hemisphere. A more robust indicator for the strengthening of the dynamical driven influence of the volcanic eruption is the characteristic winter warming over Northern Europe (Shindell et al., 2004; Fischer et al., 2007). By averaging over the whole first year after the eruption we still could find a significant positive signal in Northern Europe due to the winter

warming. This means that the general decrease of the surface temperature due to the injected aerosols is overwhelmed by this dynamical effect at mid-latitudes. This contributes to the fact that the strength of the radiative driven cooling is still uncertain. To evaluate the cooling effect qualitatively it is necessary to separate the volcanic impact on the surface temperature from other signals. This can be done by applying regression techniques directly to observations or reanalysis products (e.g. Mitchell et al., 2014; Fujiwara et al., 2015). The disadvantage is that no noise is considered for the individual regressor variables. Instead we

used detection and attribution with all available CMIP5 simulations which just include volcanic forcing in order to separate the volcanic signal from solar or anthropogenic forcings. We showed that the tropical surface temperature response to solar forcing is not detectable in any model. The multi-model-mean shows an overestimation of the volcanic signal with a mean scaling factor of around 0.5. This means, that the cooling of the mean surface temperature is much less pronounced than suggested from the model simulations. This apparent overestimation was hypothesised to be due to aliasing of volcanoes with El Niño

events in the observations (if indeed the El Niño events are not themselves a response to volcanic eruptions), and a too strong tropical stratospheric response in models (resulting in less solar radiation reaching the surface).

*Acknowledgements.* We thank Prof Lesley Gray for insightful comments. We also thank Qiuzi Han Wen for providing the program code of the optimal fingerprint techniques and for helpful comments on the work.



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





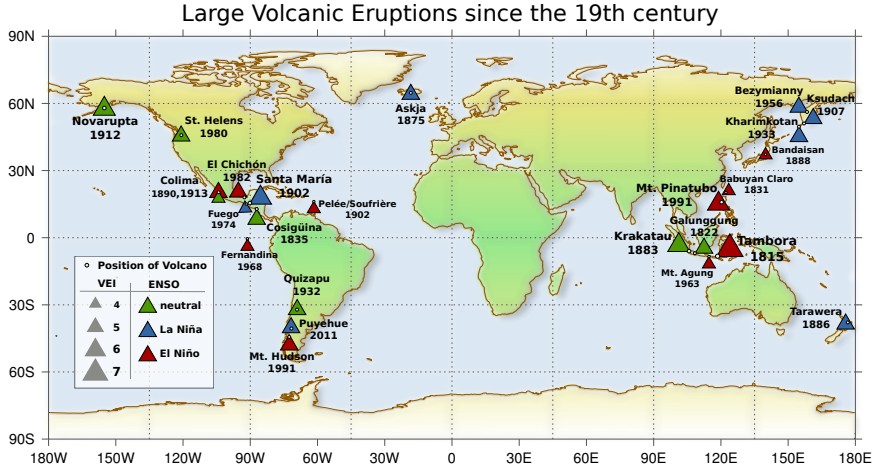

**Figure 1.** Large volcanic eruptions since the 19th century. The size of the triangle indicates the volcanic explosivity index (VEI) (Newhall and Self, 1982), from Simkin and Siebert (1994) and the Smithsonian Institution [www.volcano.si.edu]. The filled colour shows the El Niño-Southern Oscillation (ENSO) event in the first winter after the eruption from Cook et al. (2009) until 1870 and from Wolter and Timlin (2011) from 1871 until present. In the case of Fuego, Santa María, Babuyan Claro and Galunggung the second winter after the eruption is considered for the definition of the ENSO phase.

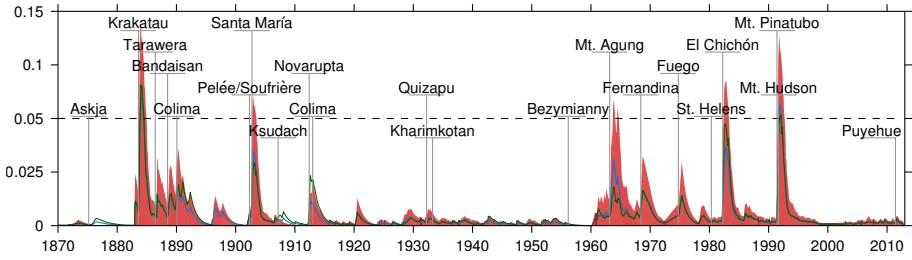

**Figure 2.** Tropical averaged Stratospheric Aerosol Optical Depth (AOD) at 550nm for the altitude region 20-25 km from Sato et al. (1993), updated dataset from data.giss.nasa.gov/modelforce/strataer/. All volcanic eruptions from Figure 1 for the period 1870 until present are included. They are indicated with a gray line corresponding to the eruption time and the name of the volcano. The green line is the mean AOD for the Northern Hemisphere and the blue line shows the mean AOD for the Southern Hemisphere.





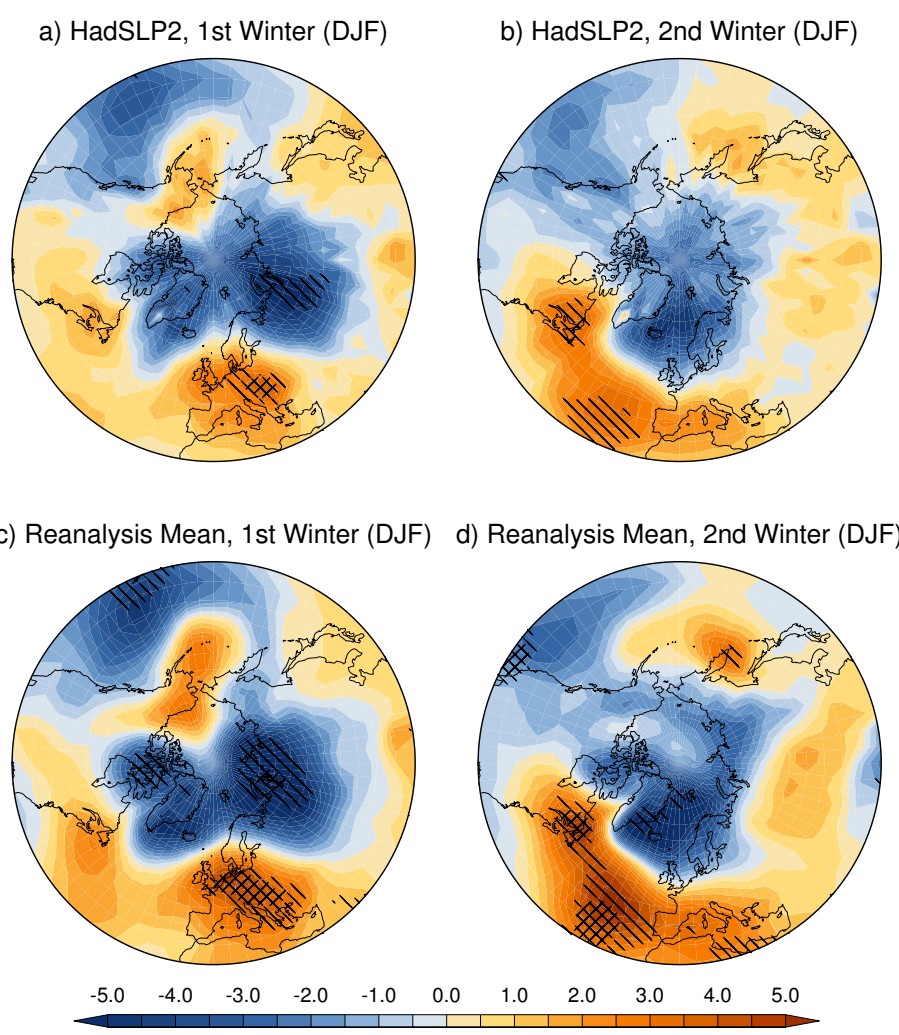

**Figure 3.** Observed and multireanalysis-mean SLP anomalies (hPa) in the Northern Hemisphere in the first two post volcanic winters (DJF) averaged over the two volcanic eruptions of Mt. Pinatubo (1991) and El Chichón (1982). Anomalies are calculated with respect to the mean for the years 1979-2012, excluding the following two years after the eruptions. Single diagonal lines correspond to the 90% and double diagonal lines to the 95% confidence level obtained with a Monte Carlo test of two independent samples. a) Shows the mean anomaly for the HadSLP observation data for the first and b) for the second winter after the eruptions. c) and d) show respectively the mean anomaly of all 10 reanalysis datasets (Table 1).





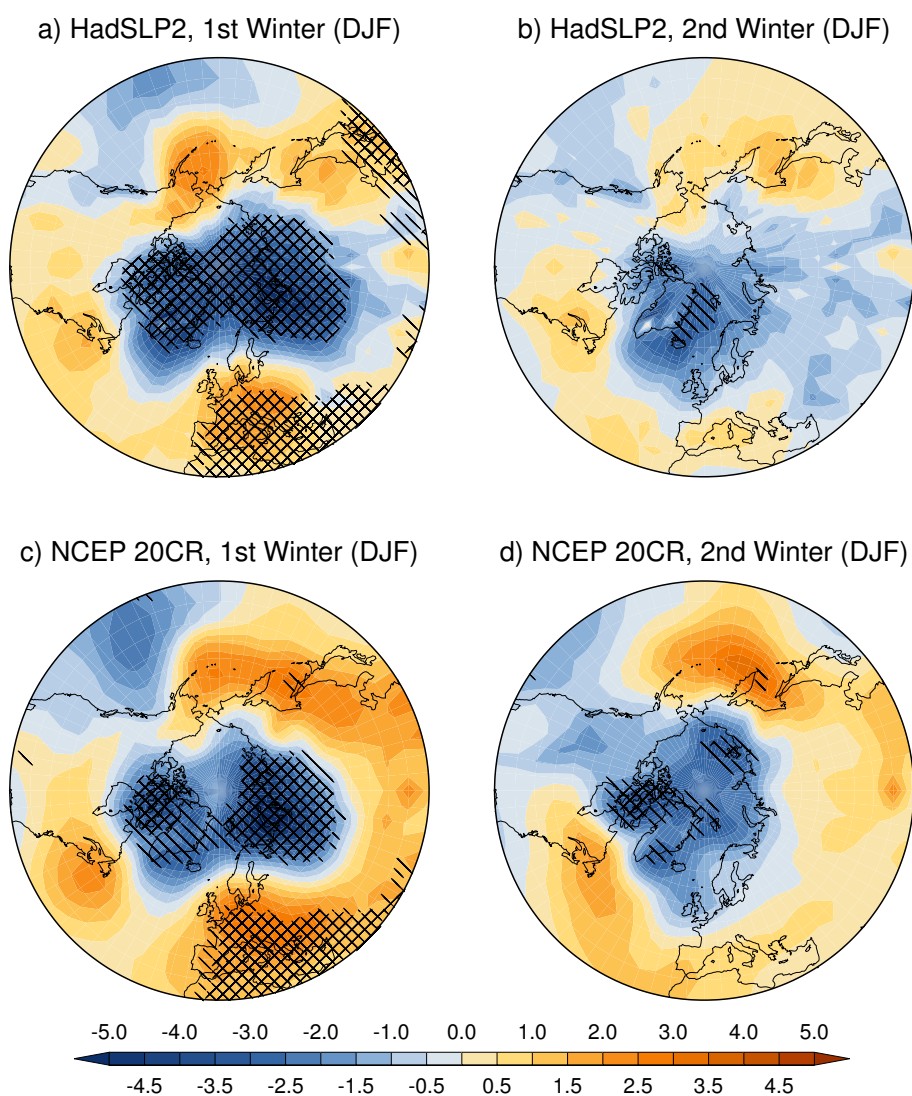

**Figure 4.** As Figure 3 but for just NCEP-20CR data and HadSLP2 data, averaged over all five post volcanic winters.




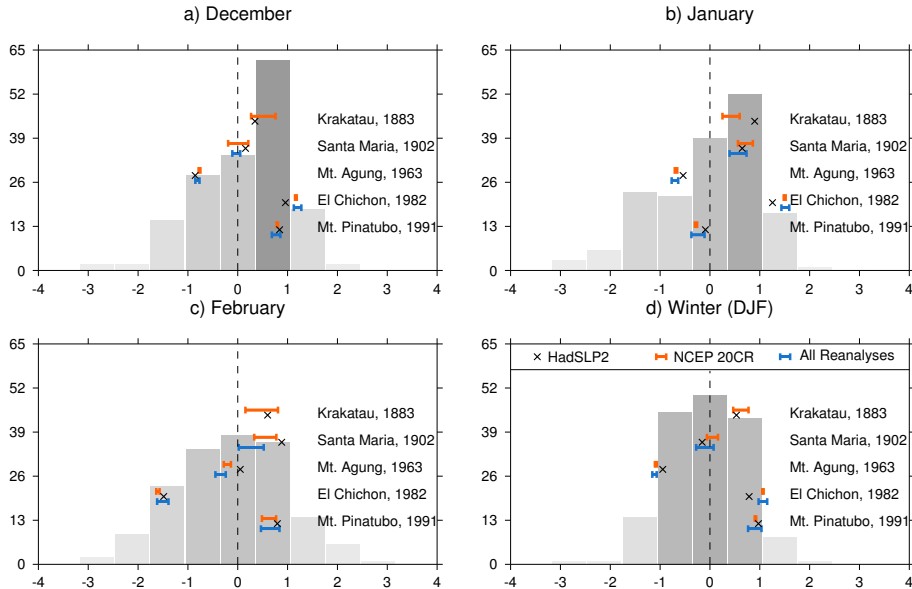

**Figure 5.** Monthly North Atlantic Oscillation (NAO) index of the first winter after five volcanic eruptions (a-c) and the winter mean (d), calculated with HadSLP2 observation data and reanalysis data. All data is calculated with respect to the mean for the years 1979-2012, excluding the following two years after the two eruption. The EOF is calculated over the period 1979-2012 for every product separately. The histogram shows the NAO index of the 163 years of observation data (1850-2012). Black crosses show the results of the HadSLP2 data. The blue lines show the reanalysis data spread of the NAO index for the winter after Mt. Pinatubo and El Chichón. The orange lines show the 95% ensemble spread of the NAO after all five volcanic eruptions calculated with the NCEP-20CR dataset.



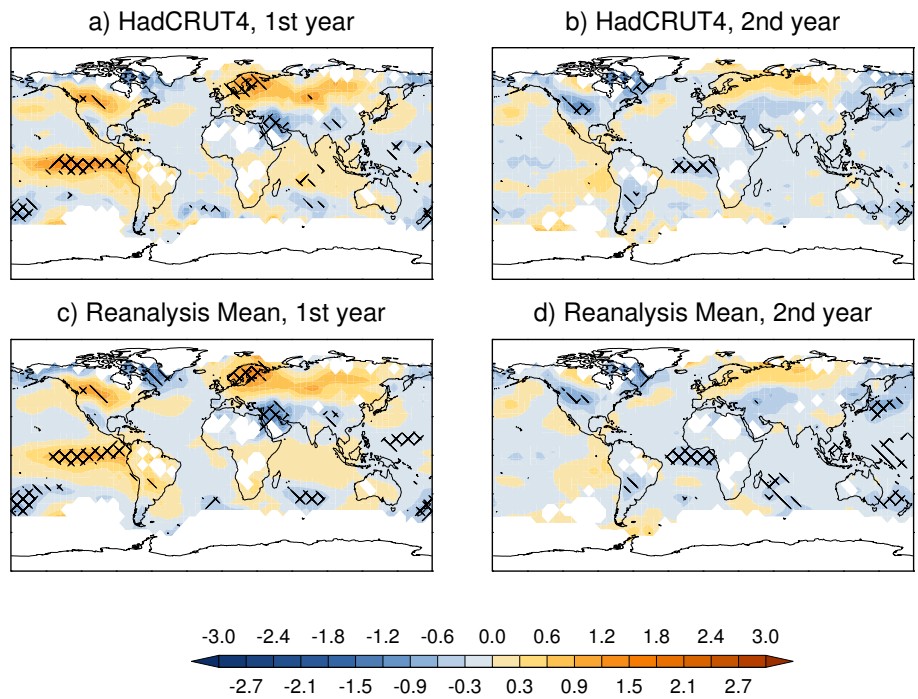

**Figure 6.** Observed and multireanalysis-mean TAS anomalies (K) averaged over the first year (a and c) and second year (b and d) after the eruption and over the three volcanic eruptions Mt. Pinatubo (1991), El Chichón (1982) and Mt. Agung (1963). a) and b) show the mean anomaly for the HadCRUT4 observations and c) and d) show respectively the mean anomaly of the five reanalysis datasets, containing this period (Table 1). Anomalies are calculated with respect to the average over the years 1961-1990, consistent to the HadCRUT4 dataset. Trends are removed with a adapted 10-year running mean. Single diagonal lines correspond to the 90% and double diagonal lines to the 95% confidence level obtained with a Monte Carlo test of two independent samples.



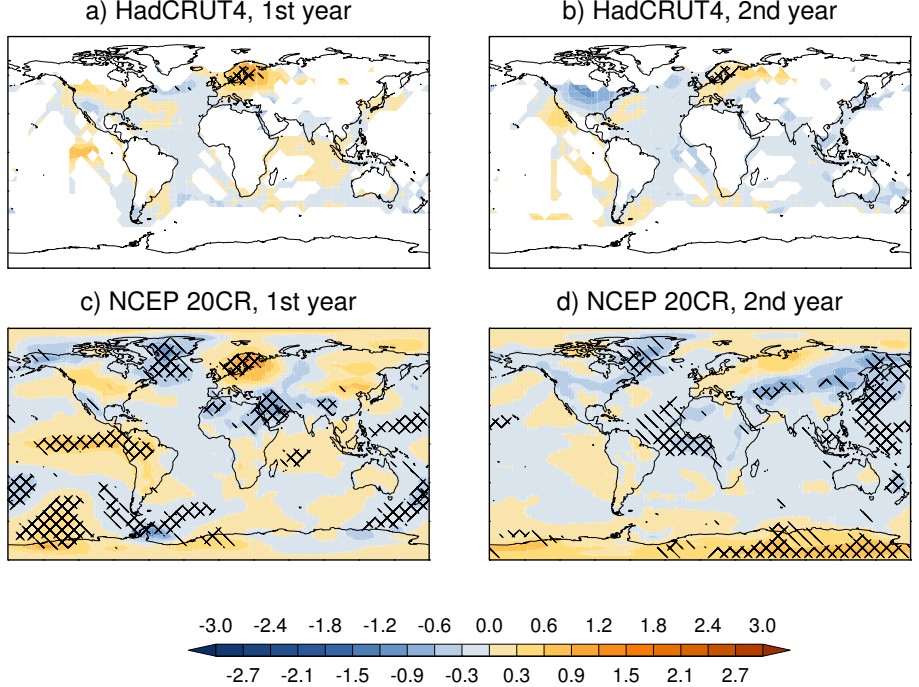

**Figure 7.** As Figure 6 but for just HadCRUT4 data in a-b averaged over at least four of five eruptions and NCEP-20CR data without missing data considereration, averaged over all five post volcanic years.

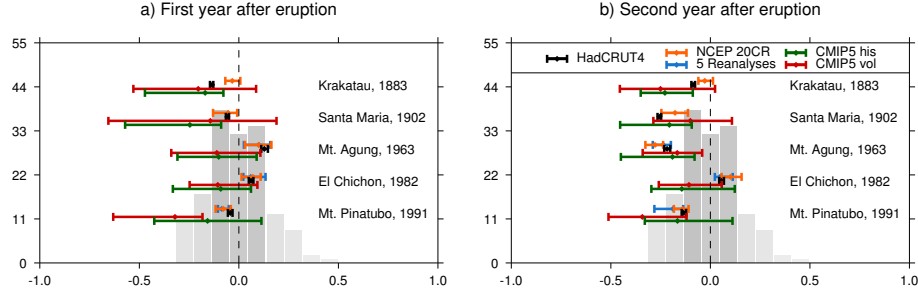

**Figure 8.** Tropical TAS anomalies (K) averaged over the region 30°S-30°N and over the first year (a) and second year (b) after five volcanic eruptions, calculated with HadCRUT4 observation data and reanalysis data. Anomalies are calculated with respect to the average over the years 1961-1990, consistent to the HadCRUT4 dataset. Trends are removed with a adapted 10-year running mean. The histogram shows the anomalies of 153 years of observation data (1855-2007). The frist and last five years are leaved out because of the trend removal. The blue lines show the reanalysis data spread of the TAS anomalies for the years after Mt. Pinatubo and El Chichón. The orange lines show the 90% ensemble spread of the NAO after all five volcanic eruptions calculated with the NCEP-20CR dataset. The mean response is indicated by the dot between the whiskers.





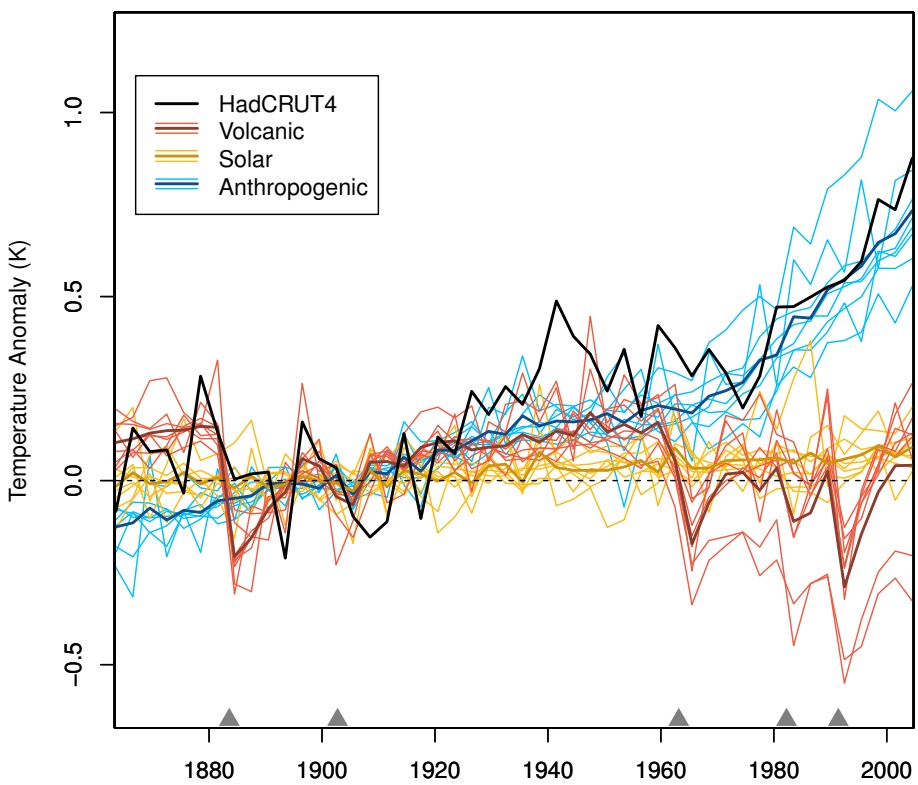

**Figure 9.** Tropical three year mean TAS variations in K (1862-2005) with respect to 1880-1919 for the CMIP5 historicalMisc experiments with just volcanic forcing in red, solar forcing in yellow and just anthropogenic forcing in blue of every model (Table 2). The black solid line shows HadCRUT4 observation data. Light color lines are ensemble means of every model and solid lines show the multimodel mean. Gray triangles indicate large tropical volcanic eruptions.




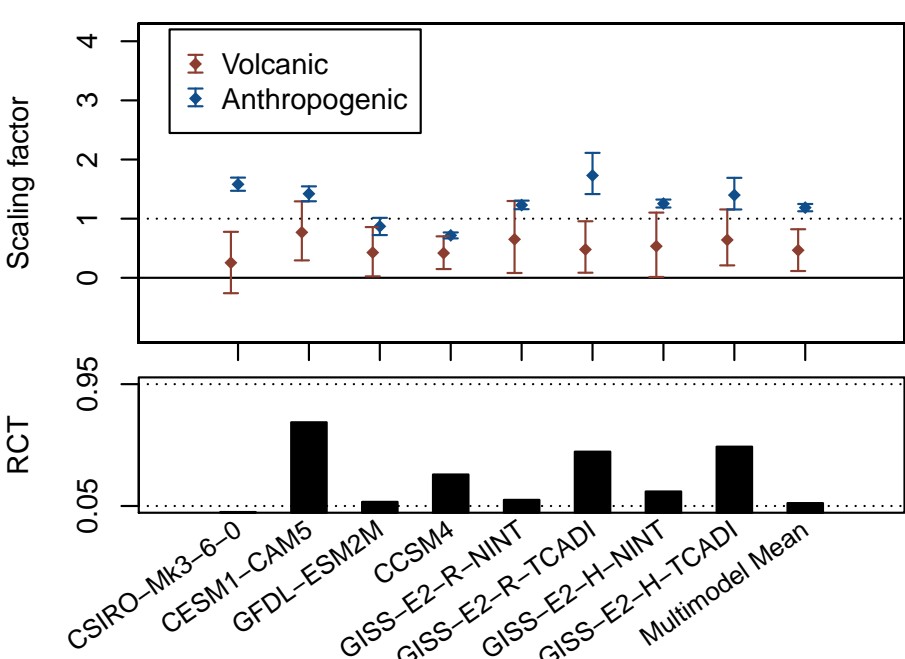

**Figure 10.** Scaling factors best-estimates (diamond) and conficence interval of tropical TAS (1862-2005) for all different CMIP5-models (Table 2) and multimodel mean calculated using the ROF method. The p values of the confidence interval, produced with a RCT, are shown in the lower panel.





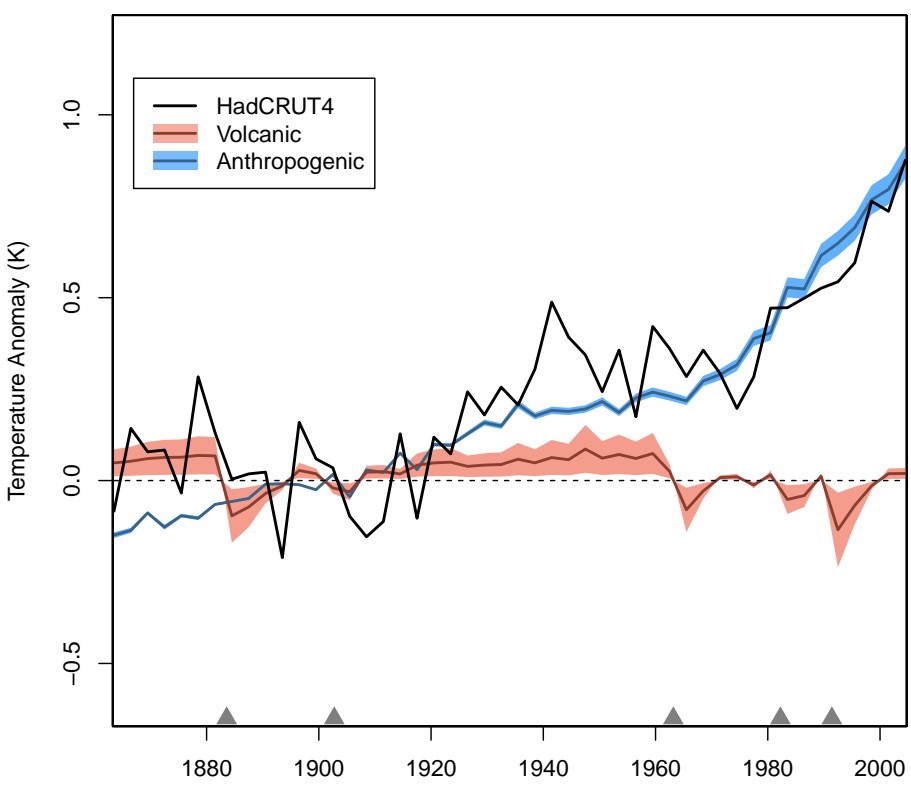

**Figure 11.** Tropical rescaled three year mean TAS variations in K (1862-2005) with respect to 1880-1919 for the CMIP5 historicalMisc experiments with just volcanic forcing in red and just anthropogenic forcing in blue. The rescaling uncertainties due to the confidence interval are shown in lightred and lightblue respectively. The black solid line shows HadCRUT4 observation data. Gray triangles indicate volcanic eruptions.





**Table 1.** Reanalysis products used in this study

| Reanalysis Name | Reanalysis Resolution | Ensemble members | Time span | Volcanic aerosol changes | Reference |
|---|---|---|---|---|---|
| NASA MERRA | 2/3° Lat x 1/2° Lon, L72 | 1 | Jan 1979 - present | no | Rienecker et al. (2011) |
| ERA-40 | T159, L60 | 1 | Sept 1957 - Aug 2002 | no | Uppala et al. (2005) |
| ERA-Interim | T255, L60 | 1 | Jan 1979 - present | no | Dee et al. (2011) |
| ERA-20C | T159, L91 | 1 | Jan 1900 - Dec 2010 | no | Poli et al. (2013) |
| JRA25/JCDAS | T106, L40 | 1 | Jan 1979 - Jan 2014 | no | Onogi et al. (2007) |
| JRA55 | T319, L60 | 1 | Jan 1958 - present | no | Ebita et al. (2011) |
| NCEP-1 (R-1) | T62, L28 | 1 | Jan 1948 - present | no | Kalnay et al. (1996) |
| NCEP-2 (R-2) | T62, L28 | 1 | Jan 1979 - near present | no | Kanamitsu et al. (2002) |
| NCEP-CFSR / CFSv2 | T382, L64 | 1 | Jan 1979 - present | yes | Saha et al. (2010) |
| NOAA-CIRES 20th Century Reanalysis | T62, L28 | 56 | Jan 1871 - Dec 2012 | yes | Compo et al. (2011) |



**Table 2.** CMIP5 models used in this study

| Model Name | Model Resolution | Ensemble members | Time span | Volcanic aerosol changes | Reference |
|---|---|---|---|---|---|
| CCSM4 | 0.9° Lat x 1.25° Lon, L26 | 3 | Jan 1850 - Dec 2005 | A07[a] | Gent et al. (2011) |
| CESM1-CAM5 | 0.9° Lat x 1.25° Lon, L30 | 3 | Jan 1850 - Dec 2005 | A07 | Meehl et al. (2013) |
| CSIRO-Mk3-6-0 | T63, L18 | 5 | Jan 1850 - Dec 2012 | S93[b] | Rotstayn et al. (2010) |
| GFDL-ESM2M | C48, L48 | 1 | Jan 1860 - Dec 2005 | S93 / S98[c] | Donner et al. (2011) |
| GISS-E2-H NINT | 2° Lat x 2.5° Lon, L40 | 5 | Jan 1850 - Dec 2005 | S93 | Schmidt et al. (2014) |
| GISS-E2-H TCADI | 2° Lat x 2.5° Lon, L40 | 5 | Jan 1850 - Dec 2012 | S93 | Schmidt et al. (2014) |
| GISS-E2-R NINT | 2° Lat x 2.5° Lon, L40 | 5 | Jan 1850 - Dec 2005 | S93 | Schmidt et al. (2014) |
| GISS-E2-R TCADI | 2° Lat x 2.5° Lon, L40 | 5 | Jan 1850 Dec 2012 | S93 | Schmidt et al. (2014) |

[a] Ammann et al. (2007), [b] Sato et al. (1993), [c]Stenchikov et al. (1998).