# Peer review of "Uncertainty and detectability of climate surface response to large volcanic eruptions"

_Atmospheric Chemistry and Physics, 2016_

## Short Comment (SC1)

**Short Comment on Wunderlich and Mitchell "Uncertainty and detectability of climate surface response to large volcanic eruptions" by Karsten Haustein**

The manuscript is very interesting and deserves publication without a doubt. Identifying and disentangling the effects of volcanic eruptions on mid-latitude dynamics is a long-standing issue and the authors contribute in that they use reanalysis data to identify attributable signals in different regions at different lags. They then aim to scale the model response according to results of a regularized optimal fingerprint technique.

While the authors mention the potentially significant effect that ENSO can have for their analysis, an appropriate test of the ENSO impact has not been undertaken. Recent literature findings (Lehner et al 2016) suggest, however, that it is crucial to account for the effects of ENSO. This is particularly true as volcanic eruptions may well force ENSO into an El Nino state (e.g. Pausata et al. 2015) as briefly mentioned by the authors themselves as well. One of their main conclusions is, quote: *"The multi-model-mean shows an overestimation of the volcanic signal with a mean scaling factor of around 0.5. This means, that the cooling of the mean surface temperature is much less pronounced than suggested from the model simulations."*

Failing to prove that the removal of ENSO won't have an effect on the scaling factor is something I find hardly reconcilable with the abovementioned statement. I would therefore ask the authors re-evaluate the scaling result in the light of newer results that have been published in the meantime.

I would like to provide some support for Lehner et al. 2016 by highlighting one of my own results that I produced recently (Figure 1). I used HadCRUT4-CW (Cowtan and Way 2014) and Berkeley Earth GMST, regressed onto the Multivariate ENSO Index (MEI), volcanic AOD (Crowley and Untermann 2013) and Sunspot Numbers (SSN). Four different 30-40 year overlapping intervals have been chosen in order to select periods where there is little change in trend in the anthropogenic forcing. This way, it is ensured that the signal is effectively removed. The volcanic contribution is then removed from GMST and compared with PMIP3 and CMIP5 data for the 1850-2015 period (red, pink, black and blue line in Figure 1). CMIP5 models are selected to match the reduced number of PMIP3 models. Using all CMIP5 models instead does not alter the result.

With the exception of Krakatoa in 1883, all other major eruptions (Agung, El Chichon, Pinatubo) are well captured in terms of magnitude in the models. I would therefore argue that a scaling factor of 0.5 can hardly be reconciled with these results. Again, it appears appropriate to repeat some of the tests with carefully corrected GMST data. Having done that, the paper conclusions should be much more robust than they appear to be at the moment.

[Figure]

**Figure 1**: *Top panel: ENSO-corrected annual mean HadCRUT4-CW (red) and Berkeley Earth GMST (pink) timeseries (1850-2015) vs PMIP3 (black) and CMIP5 (blue) annual mean TAS. The 2015 mean temperature for HadCRUT4-CW, Berkeley Earth and GISS (yellow) is provided as a dot on the upper right hand side. The yellow shaded area refers to a range of sensitivities for a two-box Energy Balance Model (EBM) with the best estimate in brown. Bottom panel: Long-term volcanic response due to slow readjustment of the deep-ocean in model (HadCM3; purple) and EBM (brown).*

**References**

Crowley and Untermann, 2013: Technical details concerning development of a 1200 yr proxy index for global volcanism. doi:10.5194/essd-5-187-2013

Cowtan and Way, 2014: Coverage bias in the HadCRUT4 temperature series and its impact on recent temperature trends. doi:10.1002/qj.2297

Pausata et al. 2015: Impacts of high-latitude volcanic eruptions on ENSO and AMOC. doi:10.1073/pnas.1509153112

Lehner et al. 2016: The importance of ENSO phase during volcanic eruptions for detection and attribution. doi:10.1002/2016GL067935

---

## Referee Comment (RC1) · Anonymous Referee #1 · 14 Apr 2016

The paper investigates the impact of major volcanic eruptions on the surface temperature and pressure. Both the impact in the tropics (mainly radiative cooling) and in the extra-tropics (mainly dynamical winter warming) are considered. The paper compares the impact in 10 different reanalysis datasets and observations from HadCRUT and HadSLP. The main difference between this paper and previous work seems to be the many different reanalyses used here. The volcanic impact is at the edge of significance and in the tropics the authors use a fingerprint method to identify a signal. It is an important topic but I wonder a little how many novel results the paper contains.

Major comments:

The main new contribution seems to be the inclusion of the many reanalyses. However,

[Figure]

I doubt how much additional information the reanalyses provide when we already have the observations of both surface temperature and pressure. If the models do not get the impact of the eruptions right then why should the reanalysis products which are based on models be better than the observations?

The Introduction is very brief. I think the authors need to discuss the difference between the tropical radiative response and the extra-tropical dynamical response already here. There should also be a more detailed discussion of previous work in these two separate areas.

Minor comments:

page 4, l24: Is Adams et al. 2006 the right reference? I briefly browsed the article and couldn't find anything about volcanic eruptions. Perhaps it should be Adams et al. 2003. This paper is cited in Christiansen 2008 which discuss the subject in some detail.

Fig. 5: I think this figure is hard to understand.It is said in the caption that the blue lines are for the winter after Pinatubo and El Chichon, but there seems to be more than two blue lines in the plot. I also wonder what the histogram tells us. Should it indicate the significance? But it is not for the same source as the other data so how can they be compared?

Fig. 8: The same here. Also, now it is said that the orange curves show the NAO signal. Should it be the TAS?

page 7, l23: It would be interesting to see results when more weaker eruptions are included.

page 8: It should be mentioned in the beginning of section 3.2 that this deals with the annual mean response.

page 9, l25. If the solar signal does not add anything why not begin the discussion wit the two-signal ROF?

page 10, lines 3-12: The discussion of the sampling of El-Nino events is unclear.

page 11: l14: Is there a cooling signal in Europe in summer?

Section 2.4: The description of the fingerprint method is very brief and impossible to understand without reading the references. In this way this analysis is different from the rest of the paper. Perhaps the fingerprint analysis could be deleted?

---

## Referee Comment (RC2) · Anonymous Referee #2 · 26 Apr 2016

This paper revisits the important topic of the effects of volcanic variability to surface climate. The article begins by comparing the response to volcanic eruptions in the first and second year after the eruption date in a wide range of reanalyses and observations. The second part involves a detection and attribution method to estimate the response to volcanic eruptions in observed and modelled tropical mean temperature. The results support a stronger response to volcanic eruptions in models than in observations. The analysis seems rigorous and the results well explained.

However the results as presented do not appear sufficiently novel. More needs to be done to place them in the context of previous findings, such as those by Stenchikov et al, Christiansen 2008, Hegel et al 2011, Driscoll et al 2012 (Driscoll et al 2012 is not

referenced, if this is the paper you are referring to by D12 this should be stated and the paper referenced), many of which have gone further than this study to also compare the response to models. Given that there is already this large body of literature looking at this topic it is essential that the authors highlight any differences and improvements on previous work to make this paper relevant and worthy of publication. For example in the abstract it says that you "conclude that [the NAO] is not as clear cut as current literature suggests". This should be explored further. The introduction should therefore be greatly expanded to motivate this particular study and original findings highlighted.

In particular I do not see what more the detection and attribution results add to the large body of literature which has already reached similar conclusions with the same models and observations and very similar techniques, papers such as Ribes et al 2013, Jones et al 2013, Gillet et al 2013 all of which seem to show very similar results to those in figure 10. Although these results use natural and not volcanic simulations, given the small response in models to solar forcing, the fact that the results look very similar seems unremarkable. Therefore I think that more should be done to highlight any differences or this section should be removed and previous studies cited instead.

Minor comments:

Lehner et al 2016, have conducted a similar study analysing the effect of ENSO on detection and attribution results. Since a possible ENSO bias is mentioned throughout this article a discussion of the results found in Lehner et al 2016 should be included.

In the methods more details should be added to the meaning of the RCT test, since as it stands it is difficult to interpret the lower panel of figure 10.

Why are the anomalies with respect to 1880-1919 on figure 11?

---

## Author Comment (AC1) · 3 Aug 2016

We thank the reviewer for the in-depth assessment of our paper. The manuscript has been revised accordingly, with most points being taken into account as per the reviewer suggestions. In particular, we emphasized the main results of the paper compared to previous studies and we included a technique to remove the impact of ENSO to support the robustness of our results. We excluded the fingerprint analysis to get a consistent and clearer picture of our results. We therefore changed the title of the paper to: "Revisiting the observed climate surface response to large volcanic eruptions".

[Figure]

**Major comments:**

- The main new contribution seems to be the inclusion of the many reanalyses. However, I doubt how much additional information the reanalyses provide when we already have the observations of both surface temperature and pressure. If the models do not get the impact of the eruptions right then why should the re-analysis products which are based on models be better than the observations?

One major aim of the paper is to measure the uncertainty in the reanalysis products. This kind of direct comparison of the volcanic response between all reanalysis prod-ucts and the observations has never been done before and will contribute to the S-RIP report. Reanalyses are used to build a complete picture of the atmospheric (and other components) system. Thereby forcing the stratospheric and tropospheric state to be in the direction of that observe during a volcano will filter through to land surfaces being better as well. So we would expect it to be better than models. Part of this paper is to identify the differences in the reanalysis so that researchers who use them know which ones to use, and which to avoid. We also provide a systematic comparison of reanal-ysis, obs and models looking at both radiative and dynamical response. By revisiting the widely accepted view of the dynamical and radiative response, we conclude that they are not as robust as often stated and show that identifying the effect of volcanic eruptions is still an issue.

- The Introduction is very brief. I think the authors need to discuss the differ-ence between the tropical radiative response and the extra-tropical dynamical response already here. There should also be a more detailed discussion of pre-vious work in these two separate areas.

The introduction was adapted to include the difference between radiative and dynami-cal response to volcanic eruptions. Also added a discussion of the novelty of the paper already in the introduction.
* * *
Interactive
comment

**Minor comments**

- page 4, l24: Is Adams et al. 2006 the right reference?

Changed to Adams et al., 2003

- Fig. 5: I think this figure is hard to understand.It is said in the caption that the blue lines are for the winter after Pinatubo and El Chichon, but there seems to be more than two blue lines in the plot. I also wonder what the histogram tells us. Should it indicate the significance? But it is not for the same source as the other data so how can they be compared?

The caption of Figure 5 was changed. The histogram shows the distribution of the NAO index for the observations. This can not directly to the distribution of the reanalysis products but is very similar. Therefore the histogram acts as a measure of the strength of the NAO response after the eruptions.

- Fig. 8: The same here. Also, now it is said that the orange curves show the NAO signal. Should it be the TAS?

The caption of Figure 8 was changed. The distribution of TAS anomalies of the observations and reanalysis products are very similar. Therefore the histogram acts as a measure of the strength of the TAS response after the eruptions.

- page 7, l23: It would be interesting to see results when more weaker eruptions are included.

The NAO response after weaker eruptions is shown in Figure S4 of the supplement. After the eruption of Fernandina in 1968, with a similar AOD to e.g. Fuego in 1974, a strong negative NAO was found. Therefore by including weaker eruptions still we would not find a robust NAO signal.

- page 8: It should be mentioned in the beginning of section 3.2 that this deals with the annual mean response.

Done

- page 9, l25. If the solar signal does not add anything why not begin the discussion wit the two-signal ROF?

- page 10, lines 3-12: The discussion of the sampling of El-Nino events is unclear.

- Section 2.4: The description of the fingerprint method is very brief and impossible to understand without reading the references. In this way this analysis is different from the rest of the paper. Perhaps the fingerprint analysis could be deleted?

We excluded the fingerprint analysis.

- page 11: l14: Is there a cooling signal in Europe in summer?

There is a minor but in general not significant summer cooling in Europe following volcanic eruptions with a maximum over Scandinavia. This was not explicitly shown by our analysis but found by e.g. Fischer et al., 2007.

---

## Author Comment (AC3) · 3 Aug 2016

We thank Karsten Haustein for the short comment on our paper. The manuscript has been revised accordingly, with most points being taken into account as per the reviewer suggestions. According to the results of Lehner et al., 2016 we included a technique to remove the impact of ENSO to support the robustness of our results. We excluded the fingerprint analysis to get a consistent and clearer picture of our results. We therefore changed the title of the paper to: "Revisiting the observed climate surface response to large volcanic eruptions".

---

## Author Response (AR1)

**Author's Response**

**Response to Anonymous Referee #1 of "Uncertainty and detectability of climate surface response to large volcanic eruptions".**

Major comments:

- The main new contribution seems to be the inclusion of the many reanalyses. However, I doubt how much additional information the reanalyses provide when we already have the observations of both surface temperature and pressure. If the models do not get the impact of the eruptions right then why should the reanalysis products which are based on models be better than the observations?

**One major aim of the paper is to measure the uncertainty in the reanalysis products. This kind of direct comparison of the volcanic response between all reanalysis products and the observations has never been done before and will contribute to the S-RIP report. Reanalyses are used to build a complete picture of the atmospheric (and other components) system. Thereby forcing the stratospheric and tropospheric state to be in the direction of that observe during a volcano will filter through to land surfaces being better as well. So we would expect it to be better than models. Part of this paper is to identify the differences in the reanalysis so that researchers who use them know which ones to use, and which to avoid. We also provide a systematic comparison of reanalysis, obs and models looking at both radiative and dynamical response. By revisiting the widely accepted view of the dynamical and radiative response, we conclude that they are not as robust as often stated and show that identifying the effect of volcanic eruptions is still an issue.**

- The Introduction is very brief. I think the authors need to discuss the difference between the tropical radiative response and the extra-tropical dynamical response already here. There should also be a more detailed discussion of previous work in these two separate areas.

**The introduction was adapted to include the difference between radiative and dynamical response to volcanic eruptions. Also added a discussion of the novelty of the paper already in the introduction.**

Minor comments

- page 4, l24: Is Adams et al. 2006 the right reference?
**Changed to Adams et al., 2003**

- Fig. 5: I think this figure is hard to understand.It is said in the caption that the blue lines are for the winter after Pinatubo and El Chichon, but there seems to be more than two blue lines in the plot. I also wonder what the histogram tells us. Should it indicate the significance? But it is not for the same source as the other data so how can they be compared?

**The caption of Figure 5 was changed. The histrogram shows the distribution of the NAO index for the observations. This can not directly to the distribution of the reanalysis products but is very similar. Therefore the histrogram acts as a measure of the strength of the NAO response after the eruptions.**

- Fig. 8: The same here. Also, now it is said that the orange curves show the NAO signal. Should it be the TAS?

**The caption of Figure 8 was changed. The distrubution of TAS anomalies of the observations and reanalysis products are very similiar. Therefore the histrogram acts as a measure of the strength of the TAS response after the eruptions.**

- page 7, l23: It would be interesting to see results when more weaker eruptions are included.

**The NAO response after weaker eruptions is shown in Figure S4 of the supplement. After the erruption of Fernandina in 1968, with a similar AOD to e.g. Fuego in 1974, a strong negative NAO was found. Therefore by including weaker eruptions still we would not find a robust NAO signal.**

- page 8: It should be mentioned in the beginning of section 3.2 that this deals with the annual mean response.

**Done**

- page 9, l25. If the solar signal does not add anything why not begin the discussion wit the two-signal ROF?
- page 10, lines 3-12: The discussion of the sampling of El-Nino events is unclear.
- Section 2.4: The description of the fingerprint method is very brief and impossible to understand without reading the references. In this way this analysis is different from the rest of the paper. Perhaps the fingerprint analysis could be deleted?

**We excluded the fingerprint analysis.**

- page 11: l14: Is there a cooling signal in Europe in summer?

**There is a minor but in general not significant summer cooling in Europe following volcanic eruptions with a maximum over scandinavia. This was not explicity shown by our analysis but found by e.g. Fischer et al., 2007.**

**Response to Anonymous Referee #2 of "Uncertainty and detectability of climate surface response to large volcanic eruptions".**

Major comments:

- However the results as presented do not appear sufficiently novel. More needs to be done to place them in the context of previous findings, such as those by Stenchikov et al, Christiansen 2008, Hegel et al 2011, Driscoll et al 2012.

**One major aim of the paper is to measure the uncertainty in the reanalysis products. This kind of direct comparison of the volcanic response between all reanalysis products and the observations has never been done before and will contribute to the S-RIP report. Reanalyses are used to build a complete picture of the atmospheric (and other components) system. Thereby forcing the stratospheric and tropospheric state to be in the direction of that observe during a volcano will filter through to land surfaces being better as well. So we would expect it to be better than models. Part of this paper is to identify the differences in the reanalysis so that researchers who use them know which ones to use, and which to avoid. We also provide a systematic comparison of reanalysis, obs and models looking at both radiative and dynamical response. By revisiting the widely accepted view of the dynamical and radiative response, we conclude that they are not as robust as often stated and show that identifying the effect of volcanic eruptions is still an issue.**

- In particular I do not see what more the detection and attribution results add to the large body of literature which has already reached similar conclusions with the same models and observations and very similar techniques, papers such as Ribes et al 2013, Jones et al 2013, Gillet et al 2013 all of which seem to show very similar results to those in figure 10.

**We excluded the detection and attribution analysis.**

Minor comments:

- Lehner et al 2016, have conducted a similar study analysing the effect of ENSO on detection and attribution results. Since a possible ENSO bias is mentioned throughout this article a discussion of the results found in Lehner et al 2016 should be included.

- In the methods more details should be added to the meaning of the RCT test, since as it stands it is difficult to interpret the lower panel of figure 10.

- Why are the anomalies with respect to 1880-1919 on figure 11?

**We applied a ENSO removal technique to get more robust results. Since we exluded the detection and attribution analysis, the minor comments are negligible.**

Response to Karsten Haustein of "Uncertainty and detectability of climate surface response to large volcanic eruptions".

**According to the results of Lehner et al., 2016 we included a technique to remove the impact of ENSO to support the robustness of our results. We excluded the fingerprint analysis to get a consistent and clearer picture of our results. We therefore changed the title of the paper to: "Revisiting the observed climate surface response to large volcanic eruptions".**

Summary

**We thank the reviewers for the in-depth assessment of our paper. The manuscript has been revised accordingly, with most points being taken into account as per the reviewer suggestions.**

- **In particular, we emphasized the main results of the paper compared to previous studies by extending the introduction and discussion.**
- **We included a technique to remove the impact of ENSO to support the robustness of our results. Because of the more sophisticated analysis technique, we updated all Figures except Figure 1 and Figure 2.**
- **We used a newer version of the NOAA-20CR reanalysis product (v2c).**
- **We excluded the detection and attribution analysis to get a consistent and clearer picture of our results.**

[revised manuscript text omitted]

---

## Author Response (AR2)

**Author's Response**

**Anonymous Referee #1**

•*The authors should improve the Introduction. I mentioned in my first review that there should also be a more detailed discussion of previous work. The second reviewer suggested that "The introduction should therefore be greatly expanded to motivate this particular study and original findings highlighted". I don't think this is done properly. The relevant literature should be cited here and not only in the sections with results. I also find some of the text in the Introduction confusing. In line 20 the authors cite Mitchell et al. for a study that separates the signal at the surface. But then only results from the stratosphere are mentioned.*

**The Introduction was extended by a new section and more relevant literature was cited.**

• *In the Conclusion and Discussion it is not clear which results are from the present paper and which from the previous literature. The authors should be more clear here.*

**The Conclusion and Discussion was extended with a new section and additional literature citation was added to separate our results from previous clearer.**

•*Figure 1 is a good idea. But it looks terrible! The black text is impossible to read and the orange outline around the continents do not follow the filled continents.*

**The background map of Figure 1 was simplified and the text size was increased.**

•*The authors remove the ENSO signal before the analysis of the eruptions. It should be discussed how large an effect this has.*

**In section 3.3 a commend was made how large is the effect of the ENSO removal and an additional Figure was included in the Supplementary Information.**

•*The two last lines in the abstract: " .. agree well on the strength .. " and " .. no clear signal found .. " seem like a contradiction and should be either corrected or explained.*

**The last two lines of the abstract were modified to eliminate the contradiction.**

**Anonymous Referee #2**

•*The authors now remove the effect of ENSO prior to the analysis. I think this is a useful step but the authors do not comment on what effects, if any, this has on their results. A short comment should be included and perhaps a figure added to the supplement particularly if the effects are large.*

**In section 3.3 a commend was made how large is the effect of the ENSO removal and an additional Figure was included in the Supplementary Information.**

•*The authors have removed the section on detection and attribution, since it was adding little new. I agree with this change but feel that since the paper addresses the temperature response a few sentences discussing what other detection and attribution studies have found in this area would be useful.*

**A section about detection and attribution studies was added to the Introduction.**

•*In their response to reviews the authors have mentioned that part of the motivation of the study is "to identify the differences in the reanalysis so that researchers who use them know which ones to use, and which to avoid." This is an important aim and a short section should be added to the discussion/conclusions addressing this.*

**A section on this was added to the Discussion and Conclusion.**

•*Last line of the abstract – "high uncertainty of the model response" – Can you clarify why this is – is it due to internal variability or difference in response between models?*

**A sentence on this was added at the end of section 3.2 .**

•*P5 Line 6 –> change to "some studies show"*
*P 7 line 28 - > "could show" change to "show"*
*P 9 – line 5-7 – The significance of this should be mentioned.*
*P10 line 2 - "considering*

**All changes done.**

**Summary**

We thank the reviewers for the assessment of our paper. The manuscript has been revised accordingly, with the points being taken into account as per the reviewer suggestions.

- In partucular, we extended the paper by a new section in the introduction, revisiting important literature.
- We added some more results into the abstract and the conclusion section to emphasize the important contribution to the S-RIP project.
- We simplified Figure 1 to make it better readable.
- We made a commend on how large the removal of the ENSO effect is and added a Figure in the  Supplementary Information.

[revised manuscript text omitted]

---

## Author Response (AR3)

**Author's Response**

**Summary**

We thank the Co-Editor Peter Haynes for the revision of our paper. The manuscript has been revised accordingly, with the points being taken into account as per the reviewer suggestions.

- In partucular, we extended the discussion of the ENSO signal.
- The paper has been revised in terms of minor problems.
- The title changed to: **"Revisiting the observed surface climate response to large volcanic eruptions"**

**Peter Haynes**

- *My view is that you have responded adequately to all the referees' comments, except, perhaps, to that regarding the El Nino signal. Both referees made comments on the removal of the El Nino signal — to both you have responded 'In Section 3.3 a commend was made how large is the effect of the ENSO removal and an additional Figure was included in the Supplementary Information'.*
*You refer to this Figure (S5) in Section 3.2 and say 'Only little temperature effects can be attributed to ENSO in the extra tropics (Figure S5).' — but I'm not clear on the connection between Figure S5 and El Nino. There is no mention of El Nino in the caption, for example. Perhaps the point is that the means shown in this Figure do not have the El Nino signal removed and therefore show a strong warm signal in the E Pacific? Whatever the logic here, it should be clearly and explicitly explained.*

**In section 3.3 the ENSO signal was further discussed with a few more sentences. The Figures S5 and 6 show the mean over the last three large tropical eruptions. In Figure 6 the ENSO signal is removed, in Figure S5 not. Therefore the caption of Figure S5 was changed to make the differnces between this plot and Figure 6 clearer. The strong differences between the two figures show that the post volcanic years were higly impacted by ENSO.**

- *p7 l27: 'Figures S3' > 'Figure S3'*

- *p8 l17: 'is import' > 'is important'*

- *p8 l18: 'Only little' > 'Only small' would be clearer.*

**All changes done.**

- *p9 l17: You don't seem to have included a reference to Figure S6 in the text.*

**A reference of Figure S6 was added in the conclusion section.**

Further changes

[revised manuscript text omitted]